# Global PIEZO1 Gain-of-Function Mutation Causes Cardiac Hypertrophy and Fibrosis in Mice

**DOI:** 10.3390/cells11071199

**Published:** 2022-04-02

**Authors:** Fiona Bartoli, Elizabeth L. Evans, Nicola M. Blythe, Leander Stewart, Eulashini Chuntharpursat-Bon, Marjolaine Debant, Katie E. Musialowski, Laeticia Lichtenstein, Gregory Parsonage, T. Simon Futers, Neil A. Turner, David J. Beech

**Affiliations:** Leeds Institute of Cardiovascular and Metabolic Medicine, School of Medicine, University of Leeds, Leeds LS2 9JT, UK; elevans@uci.edu (E.L.E.); nicolamblythe@aol.com (N.M.B.); l.stewart@leeds.ac.uk (L.S.); medechu@leeds.ac.uk (E.C.-B.); m.debant@leeds.ac.uk (M.D.); kmusialowski@yahoo.co.uk (K.E.M.); l.lichtenstein@leeds.ac.uk (L.L.); g.parsonage@leeds.ac.uk (G.P.); t.s.futers@leeds.ac.uk (T.S.F.); n.a.turner@leeds.ac.uk (N.A.T.)

**Keywords:** PIEZO1, mechanotransduction, heart, hypertrophy, fibrosis, fibroblasts, calcium

## Abstract

PIEZO1 is a subunit of mechanically-activated, nonselective cation channels. Gain-of-function PIEZO1 mutations are associated with dehydrated hereditary stomatocytosis (DHS), a type of anaemia, due to abnormal red blood cell function. Here, we hypothesised additional effects on the heart. Consistent with this hypothesis, mice engineered to contain the M2241R mutation in PIEZO1 to mimic a DHS mutation had increased cardiac mass and interventricular septum thickness at 8–12 weeks of age, without altered cardiac contractility. Myocyte size was greater and there was increased expression of genes associated with cardiac hypertrophy (*Anp*, *Acta1* and *β-MHC*). There was also cardiac fibrosis, increased expression of *Col3a1* (a gene associated with fibrosis) and increased responses of isolated cardiac fibroblasts to PIEZO1 agonism. The data suggest detrimental effects of excess PIEZO1 activity on the heart, mediated in part by amplified PIEZO1 function in cardiac fibroblasts.

## 1. Introduction

The heart has an intrinsic ability to sense and respond to mechanical stimuli to maintain haemodynamic stability [1]. This process involves the conversion of mechanical stimuli into biochemical events that induce changes in myocardial structure and function. These downstream effects are initially adaptive compensatory responses, but prolonged mechanical stress leads to pathological hypertrophy and fibrosis, and eventually heart failure (HF) [1,2,3,4]. Stretch-activated ion channels have been proposed as mechanical sensors and transducers in the heart [5,6], and recent attention has been given to PIEZO1 calcium (Ca^2+^)-permeable, nonselective cation channels, which are activated by mechanical forces, such as membrane stretch and fluid flow [7,8,9,10,11,12]. PIEZO1 channels are expressed in cardiac fibroblasts, where they respond to membrane stretch and matrix stiffness, signalling to important downstream mediators, such as interleukin-6 (IL-6), p38 mitogen-activated protein kinase (MAPK), brain natriuretic peptide (BNP), tenascin C (TNC) and transforming growth factor β1 gene expression [13,14,15,16]. Mechanical stretch has been linked to the triggering of inflammatory cascade in the heart, notably through fibroblast activation [17,18]. PIEZO1 is emerging as a potential target in a broad range of inflammation diseases and healing responses [19,20,21,22]. However, PIEZO1-dependent inflammatory response in cardiac fibroblasts appeared to be specific to IL-6 as other proinflammatory cytokines, such as IL-1, were not dysregulated [13]. PIEZO1 is also expressed in cardiac myocytes but at a low level, and cardiac myocyte-specific disruption of PIEZO1 has little or no effect on the cardiac function of young adult mice, while older mice display spontaneous defective pump function [23]. Mice overexpressing PIEZO1 in cardiomyocytes show the same spontaneous HF phenotype, suggesting a role for PIEZO1 channel in maintaining normal heart function. Furthermore, PIEZO1 expression is upregulated during cardiac insult—i.e., in cardiac hypertrophy and HF—and the loss of cardiomyocyte PIEZO1 inhibits adverse remodelling caused by aortic banding in mice [23,24,25,26]. These data suggest the possibility of adverse cardiac effects of gain-of-function (GOF) PIEZO1 mutations that have been linked to dehydrated hereditary stomatocytosis (DHS), a type of anaemia [27,28,29,30]. The GOF mutations all apparently slow channel kinetics, leading to extended opening times and increased PIEZO1 function [28,30,31]. They cause PIEZO1 GOF in red blood cells (RBCs) and so might similarly increase PIEZO1 activity in other cell types. To explore this possibility, we generated a mouse model containing the murine equivalent (M2241R) of a human PIEZO1 GOF mutation (M2225R) [31]. These mice recapitulate many of the features of DHS, as do mice with another PIEZO1 GOF mutation [32]. Therefore, the M2241R mouse is a good model of the human DHS condition, showing the phenotype in heterozygous and homozygous states [31]. In this study, we investigated whether M2241R affects the heart in young adult mice.

## 2. Materials and Methods

The use of animal was authorised by the University of Leeds Animal Welfare and Ethical Review Committee and The Home Office, UK.

### 2.1. Genetically Modified Mice

CRISPR/Cas9 methodology was used to generate the mice harbouring the GOF M2241R mutation (PIEZO1^M-R/M-R^), as previously described [31]. PIEZO1^M-R/M-R^ mice and control wild-type (WT) mice (PIEZO1^WT/WT^) were housed as reported previously [33]. Genotyping was performed using real-time PCR with specific primers (Transnetyx Inc., Cordova, TN, USA). All experiments were conducted on 8–12-week-old homozygous for the mutation (PIEZO1^M-R/M-R^) and control (PIEZO1^WT/WT^) male mice.

### 2.2. Echocardiography

Echocardiography and analysis were performed as previously described [33]. With 2D imaging, M-mode echocardiography in short-axis view of the heart was measured over the entire cardiac cycle. The LV volumes, ejection fraction (EF%), fractional shortening (FS%) and the corrected LV mass were calculated using the Vevo LAB cardiac package software (v1.7.0, FUJIFILM VisualSonics Inc., Toronto, Canada) with the following formulas: LV vol d/s: ((7.0/(2.4 + LVIDd/s)) × LVIDd/s^3^, EF: 100 × ((LV vol d−LV vol s)/LV vol d), FS: 100 × ((LVIDd−LVIDs)/LVIDd), corrected LV mass: 0.8 × 1.053 × ((LVIDd + LVPWd + IVSd)^3^LVIDd^3^).

### 2.3. Blood Pressure Measurements

Blood pressure was measured noninvasively using the CODA tail-cuff system (Kent Scientific, Torrington, CT, USA) which uses volume pressure recording (VPR) to measure blood pressure by determining the tail blood volume. Mice were placed into a holder to restrict movement and allow access to the tail. The mice were placed into an incubator set at 36 °C and left for 15 min to thermoregulate and acclimatise to the holder. The recording sessions were conducted by the CODA software (v3.0, Kent Scientific, Torrington, CT, USA) and consisted of 20 cycles (5 of which were acclimatization cycles) with 5 s between each cycle. The maximum occlusion pressure was 250 mmHg, deflation time 20 s and minimum volume 15 μL. 2.4. Animals and Tissue Harvest

### 2.4. Animals and Tissue Harvest

Animals were euthanized in accordance with Schedule 1 Code of Practice, UK Animals Scientific Procedures Act 1986. Hearts were removed, cleaned and cut in half transversely. The distal part, used for immunohistochemistry, was first mounted in OCT (Tissue-Tek). It was then snap-frozen in liquid nitrogen–precooled isopentane. The proximal part, used for molecular biology, was directly snap-frozen in liquid nitrogen. All tissues were stored at −80 °C.

### 2.5. Tissue Processing and Immunohistochemistry

#### 2.5.1. Sectioning of Heart Muscle

Heart serial cross-sections (10 μm thickness) were obtained as previously described [33]. 

#### 2.5.2. In Situ Determination of Heart Cross-Sectional Area and Fibrosis

For immunofluorescence detection of fibrosis and cross-section area (CSA) measurements, slides were left to dry for 30 min at room temperature (RT). After rehydration with PBS, samples were fixed for 10 min with 4% PFA, washed, and permeabilized for 10 min with 0.1% triton X100. After 1 h of blocking with 1% BSA in PBS, heart sections were stained at RT for 1 h with CF^®^488A-conjugated wheat germ agglutinin (WGA) (Biotium, Fremont, CA, USA, 29022-1, 1:100) to label cell membranes and fibrotic scar tissue [34]. Slides were washed and mounted in ProLong Gold Antifade Mountant (ThermoFisher Scientific, Waltham, MA, USA, P36934). 

### 2.6. Slide Imaging and Quantification

To determine cardiomyocyte CSA and fibrosis, stained sections were imaged using a LSM710 Axio Examiner (Carl Zeiss Ltd., Cambridge, UK) confocal microscope. WGA-488 fluorescence was detected using excitation by 488 Argon laser and detector bandwidth of 498–557 nm. A combination of z-stack and tile-scan images were acquired using a Plan-apochromat 20× (0.8NA) M27 air objective. Maximum intensity images were generated from the z-stack and tile scans were stitched together. Images were exported and analysed with ImageJ software (v1.52p, Bethesda, MD, USA). The CSA was calculated from 5–10 random fields per animal using the Cross-Sectional Analyzer plugin. The fibrosis quantification was performed on whole sections and expressed as a percentage of total tissue. 

### 2.7. Murine Cardiac Fibroblast Isolation and Culture

Cardiac fibroblasts were isolated from hearts digested with collagenase. Fibroblasts were maintained in DMEM supplemented with 10% foetal calf serum and kept in an incubator at 37 °C with 5% CO_2_, as described previously [35]. Fibroblasts were placed in serum-free media 16 h before treatments with Yoda1 or Vehicle (DMSO). All experiments were performed between passage 1–2.

### 2.8. Quantitative RT-PCR of Cardiac Fibroblasts

Cardiac fibroblasts were treated for 6 h with Yoda1 (10 μM) or its vehicle (DMSO), and RNA was isolated using Aurum RNA Extraction Kit (Bio-Rad, Hercules, CA, USA). A total of 0.5 μg of RNA and random hexamer primers were used for retrotranscription using the RT system, according to the manufacturer’s instructions (Promega, Madison, WI, USA). Quantitative determination of mRNA expression levels was performed on ABI-7500 system using specific TaqMan probes (ThermoFisher Scientific, Waltham, MA, USA): mouse IL-6 (Mm00446190_m1), mouse Tnc (Mm00495662_m1) and mouse Gapdh (Mm99999915_g1) as endogenous control. Samples were compared using the comparative CT method: fold changes were calculated with the 2^−ΔΔCT^ formula.

### 2.9. Western Blotting

Cardiac fibroblasts were treated for 10 min (for p38α phosphorylation) or 6 h (for TNC expression) with Yoda1 (10 μM) or its vehicle (DMSO) and lysed in a buffer containing: 10 mM Tris (pH 7.5), 150 mM NaCl, 0.5 mM EDTA, 0.5% NP-40, protease inhibitors (Roche, Basel, Switzerland), and phosphatase inhibitors (Roche, Basel, Switzerland). A total of 25 μg of protein extracts were loaded on 10% polyacrylamide precast gel (Bio-Rad, Hercules, CA, USA). Proteins were transferred onto PVDF membranes, blocked in milk (5%) for 1 h before incubation with the primary antibody against phospho-p38 MAPK (9215, Cell Signalling Technology, Danvers, MA, USA; 1:250), or reprobed for p38α antibody (9228, Cell Signalling Technology, Danvers, MA, USA; 1:500) overnight at 4 °C. For TNC, membranes were incubated with primary antibody against TNC (JP10337, Tecan, Männedorf, Switzerland; 1:200) or reprobed for α-tubulin antibody (3873, Cell Signalling Technology, Danvers, MA, USA; 1:2000). Membranes were then washed and incubated with anti-mouse and anti-rabbit secondary antibodies (GE Healthcare, Chicago, IL, USA; 1:5000). Visualization was performed using ECL detection reagent and Syngene G:BOX Chemi XT4 system. Blots were analysed with ImageJ software (v1.52p, Bethesda, MD, USA). 

### 2.10. Intracellular Ca^2+^ Measurements

Cardiac fibroblasts were plated at 90% confluence in clear 96-well plates 24 h before experiments. Cells were incubated in standard bath solution (SBS containing in mM: 130 NaCl, 5 KCl, 8 D-glucose, 10 HEPES, 1.2 MgCl_2_, 1.5 CaCl_2_, pH 7.4) supplemented with 2 μM fura-2-AM (Molecular Probes™) and 0.01% pluronic acid (ThermoFisher Scientific, Waltham, MA, USA) for 1 h at 37 °C. Fibroblasts were then placed in fresh SBS at RT for 30 min to allow de-esterification of the dye. Measurements were made at RT on FlexStation III plate reader (Molecular Devices, San Jose, CA, USA), controlled by Softmax Pro software v7.0.3. The change (Δ) in intracellular Ca^2+^ was indicated as the ratio of fura-2 emission (510 nm) intensities at 340 and 380 nm excitation. 

### 2.11. RNA Extraction and Quantitative RT-PCR of Heart Samples

Total RNA from heart was extracted using TRIzol (Sigma, St. Louis, MO, USA, T9424) according to the manufacturer’s instructions. iScript cDNA Synthesis Kit (Bio-Rad, Hercules, CA, USA, 1708891) was used to synthesise cDNA from 1 μg of RNA. Determination of mRNA expression levels was conducted on a LightCycler 480 Real Time PCR System (Roche, Basel, Switzerland) using either gene-specific primers or Rpl32 gene primers as endogenous controls from Sigma (Table 1), and Sybr Green supermix (Bio-Rad, Hercules, CA, USA, 1725121). Samples were analysed using the 2^−ΔΔCT^ formula. 

### 2.12. Statistics

All data representations are mean ± standard deviation (S.D.). Figure legends state the number of mice (*n*) analysed for each experiment. Outliers were excluded according to the results of ROUT test (Q = 1%) performed in GraphPad Prism software (v9.0, San Diego, CA, USA). Figure legends also mention the statistical test used to determine statistical significance: two groups were compared using unpaired Student’s *t*-test, whereas multiple comparisons were assessed using ANOVA followed by post hoc Tukey’s test. For all experiments, significance was considered for values of *p* < 0.05. For PCR analysis, each sample was tested in duplicate. Echocardiography was performed in a blinded manner. 

## 3. Results

### 3.1. Increased Heart Size

The 8–12-week-old PIEZO1^M-R/M-R^ and matched control PIEZO1^WT/WT^ male mice were studied for cardiac structure and function assessment at the early stages of adult maturity. While the body weight of PIEZO1^M-R/M-R^ mice is similar to control mice PIEZO1^WT/WT^ (Figure 1a), increased heart mass (Figure 1b,c), heart weight to body weight ratio (Figure 1d) and heart weight to tibia length ratio (Figure 1e) were found in PIEZO1^M-R/M-R^ mice compared with PIEZO1^WT/WT^. Lung weight to body weight ratio (Figure 1f) was unaffected in PIEZO1^M-R/M-R^ mice. A potential consequence of increased PIEZO1 function could be a change in blood pressure. This parameter was measured in conscious mice and no changes were observed in mean systolic (Figure 1g), mean diastolic (Figure 1h), nor in mean arterial blood pressure (Figure 1i). The data suggest PIEZO1^M-R/M-R^ is associated with increased heart size, in the absence of elevated blood pressure.

### 3.2. Unaffected In Vivo Cardiac Contractility despite Hypertrophy

The cardiac function of PIEZO1^M-R/M-R^ mice was assessed by echocardiography. PIEZO1^M-R/M-R^ mice show signs of cardiac hypertrophy with increased left ventricular (LV) mass (Figure 2a,b) and interventricular septum thickness in systole and diastole (Figure 2c,d). LV internal diameter at end of systole (LVIDs; Figure 2e) and at end of diastole (LVIDd; Figure 2f), LV posterior wall thickness at end of systole (LVPWs; Figure 2g) and at end of diastole (LVPWd; Figure 2h), ejection fraction (Figure 2i), fractional shortening (Figure 2j), LV volumes (Figure 2k,l) and heart rates (Figure 2m) of PIEZO1^WT/WT^ and PIEZO1^M-R/M-R^ mice were similar. The data suggest that PIEZO1^M-R/M-R^ induces cardiac hypertrophy without altering cardiac contractile performance. 

### 3.3. Cellular Hypertrophy, Altered Gene Expression and Fibrosis

To investigate the underlying mechanisms of cardiac hypertrophy, we quantified cardiac myocyte size and myocardial fibrosis in heart sections stained for wheat germ agglutinin (WGA) to label cell membranes and fibrotic scar tissue (Figure 3a). PIEZO1^M-R/M-R^ mice show increased fibrosis (Figure 3b). Mean cardiomyocyte cross-sectional area was also increased (Figure 3c), indicating hypertrophy. We then quantified the expression of genes which are indicators of cardiac change in whole heart samples (Figure 3d–m): atrial natriuretic peptide (*Anp*); brain natriuretic peptide (*Bnp*); α-actin (*Acta1*); α- and β- myosin heavy chains (*α-MHC* and *β-MHC*); collagen α-1 chains I and III (*Col1a1* and *Col3a1*); connective tissue growth factor (*Ctgf*); sarco/endoplasmic reticulum Ca^2+^-ATPase 2A (*Serca2a*); PIEZO1 (*Piezo1*) (gene names are in parentheses). *Anp*, *Acta1*, *β-MHC* and *Col3a1* mRNA expression were upregulated in PIEZO1^M-R/M-R^ mice. The data suggest that the increased heart size and hypertrophy are accompanied by increased myocyte size, fibrosis and induction of hypertrophic and fibrotic factors. 

### 3.4. Increased PIEZO1 Function and Downstream Signalling in Cardiac Fibroblasts

To investigate whether the GOF mutation alters PIEZO1 function in cells of the heart itself, we isolated cardiac fibroblasts from PIEZO1^WT/WT^ and PIEZO1^M-R/M-R^ mice, and measured intracellular Ca^2+^ responses to a PIEZO1 agonist, Yoda1, as described previously [13]. The Yoda1 response is amplified in PIEZO1^M-R/M-R^ heart fibroblasts compared with PIEZO1^WT/WT^ (Figure 4a,b). Adenosine triphosphate (ATP)-evoked Ca^2+^ events were unchanged by the mutation (Figure 4c,d). PIEZO1 activity in cardiac fibroblasts prompted downstream activation of p38 MAPKα (p38-α/MAPK14) by phosphorylation and increased IL-6 secretion due to an increased expression of the *Il-6* gene [13]. Yoda1 stimulated PIEZO1^M-R/M-R^ fibroblasts displayed upregulation of *Il-6* mRNA expression (Figure 4e) and an increase in phosphorylated p38α (p-p38α) (Figure 4f,g). We then investigated the expression of TNC, a profibrotic protein known to be associated with PIEZO1 channel and IL-6 pathway [15,36]. PIEZO1^M-R/M-R^ fibroblasts showed an increased mRNA (Figure 4h) and protein expression (Figure 4i,j) of TNC upon Yoda1 treatment. The data suggest that PIEZO1 function and downstream signalling are amplified in fibroblasts of PIEZO1^M-R/M-R^ hearts.

## 4. Discussion

Our results suggest that a GOF PIEZO1 mutation causes pathological cardiac remodelling with activation of hypertrophic and fibrotic signals in the myocardium. The phenotype is correlated with amplified PIEZO1 channel function and downstream p38, IL-6 and TNC signalling in cardiac fibroblasts (Figure 5).

Hypertrophy was only detected in the septum while the posterior wall size was unchanged. This asymmetric septal hypertrophy is a major feature of human hypertrophic cardiomyopathy which is characterised by cardiomyocyte hypertrophy without extensive fibrotic changes, and is associated with preserved ventricular function [37,38,39,40]. This phenotype has been noted in early stages of pathological LV hypertrophy, while late stages are associated with global LV dysfunction [41]. 

The molecular changes observed during pathological hypertrophy are often described as a reactivation of a foetal gene program [42], with re-expression of genes primarily expressed during development. ANP and BNP are downregulated in ventricles after birth but their levels strongly increase during cardiac hypertrophy and HF [42]. Here, we observed an increase in *ANP* mRNA levels due to PIEZO1 GOF, but not *BNP* mRNA levels. While some studies show synchronous elevation of plasma ANP and BNP during volume or pressure overload and hypertension [43], others describe unchanged *BNP* mRNA levels in HF and hypertension [44]. The gene expression of *BNP* is distinctly regulated from that of *ANP* [44,45], consistent with Sakata and colleagues’ observation that *BNP* mRNA levels were unaltered at the compensatory hypertrophic stage when elevated *ANP* mRNA levels were detectable [46]. *BNP* mRNA levels increased during the transition towards HF, in the late-stage of hypertrophy and fibrosis [43,46]. Clinically, BNP testing is recommended to detect or rule out HF as increased levels are associated with LV dysfunction [47]. Although PIEZO1 has been identified as responsible for stretch-induced *BNP* expression in cardiac fibroblasts [15], fibroblasts are not the main source of myocardial BNP expression. This may explain why we did not detect a significant increase in *BNP* mRNA expression in whole heart samples. Furthermore, we do not show any ventricular dysfunction in our model, consistent with unchanged *BNP* mRNA levels. Like natriuretic peptides, β-MHC is also strongly expressed in foetal ventricles and is downregulated after birth, leaving α-MHC the major isoform in adult heart [48]. During cardiac hypertrophy, relative expression of the two isoforms is reversed, consistent with the increased *β-MHC* mRNA expression we observed in our model of the human M2225R PIEZO1 mutation. 

Cardiac fibrosis occurs as a result of cardiac fibroblast activation by diverse pathways, differentiation into myofibroblasts, together with excessive deposition of extracellular matrix proteins, in particular type I (*Col1a1*) and type III (*Col3a1*) collagens [49]. We only detected an increase in *Col3a1* gene expression in our model. However, differential regulation of *Col1a1* and *Col3a1* expressions has been reported previously, with cardiac fibroblasts subjected to mechanical load increasing *Col3a1* but not *Col1a1* mRNA levels [50]. Furthermore, substrate stiffness and stretch also differentially regulate *Col1a1* and *Col3a1* expressions in fibroblasts [4]. In patients with dilated cardiomyopathy, the myocardial ratio of collagen type III/I mRNAs was higher than in normal heart tissue. This transient inverted expression of collagen types appears to be characteristic of early stages of myocardial infarction or hypertensive myocardial fibrosis, with an initial deposition of type III collagen, followed by type I [51,52,53]. Additionally, gene expression profiling of hypertrophic cardiomyocytes revealed an upregulation of *Col3a1* expression in early-phase hypertrophy [54]. This is consistent with the modest cardiac phenotype observed in PIEZO1^M-R/M-R^ mice. Collagen content and percentage of myocardial fibrosis strongly correlate with cardiac function [53,55], with fibrosis being a major cause of diastolic dysfunction in patients with hypertrophic cardiomyopathy and HF [56,57,58,59,60]. Similarly, in several animal models of HF, cardiac dysfunction correlates with extensive fibrosis (around 10% fibrosis in myocardium), while fibrosis below 4–5% is considered to be the normal range for healthy hearts [61,62,63]. The degree of fibrosis in our model remains relatively low and this could explain why cardiac function is preserved despite the presence of pathological remodelling. This suggests that the onset of cardiac fibrosis and hypertrophy is observed from the age of 8 weeks in PIEZO1^M-R/M-R^ mice. 

We show increased Yoda1-induced Ca^2+^ entry in the cardiac fibroblasts of PIEZO1^M-R/M-R^ mice compared to those of their WT littermates. This elevated Ca^2+^ entry upon PIEZO1 activation is correlated with an increased *Il-6* gene expression, p38 phosphorylation and TNC mRNA and protein expression, consistent with previous findings from our group [13,15,35]. Furthermore, PIEZO1-mediated Ca^2+^ influx is involved in inflammatory pathways including the IL-6 receptor family and Ca^2+^-sensitive MAPK family such as p38 in various systems (e.g., renal fibrosis models, liver carcinoma cell lines or human dermal fibroblasts) [13,14,17,64,65,66,67]. IL-6 has been characterised as a key regulator of inflammation [68,69,70] and described as cardioprotective in the initial inflammatory response to cardiac pathology [71]. However, chronic elevation of IL-6 within the heart has been associated with cardiac disease progression and is known to promote cardiac fibrosis, hypertrophy and ventricular dysfunction [70,71,72,73]. Its expression is tightly regulated, with low levels of expression in healthy hearts but elevated expression during stress [74] with cardiac fibroblasts strongly synthesizing IL-6 in response to catecholamines—a well-known hypertrophic stimulus [75]. We found increased *Il-6* expression in fibroblasts upon PIEZO1 activation but could not detect it in whole heart samples. This is likely due to low basal expression and the cellular constitution of myocardium: cardiac fibroblasts being the largest cell population in terms of cell number but representing a minor proportion of total tissue volume [76]. We previously showed that PIEZO1 was not coupled to a general inflammatory response as IL-1 was not modulated by Yoda1 treatment [13]. However, PIEZO1 has been identified as responsible for stretch-induced *Bnp* and *Tgfβ1* expression in cardiac fibroblasts [15]. As we did not observe modified *Bnp* expression in our model, we hypothesise that the PIEZO1 downstream signalling pathway was specific to IL-6, but we cannot exclude the participation of other inflammatory markers, such as Tgfβ.

p38 MAPK is also known to play a role in cardiac pathological remodelling. The gene encoding p38α MAPK is required to mediate fibroblast activation in injured heart, and increased activation of p38α within fibroblasts triggers fibrosis [77]. Furthermore, cardiac fibroblast p38α contributes to cardiomyocyte hypertrophy via a fibroblast-to-cardiomyocyte IL-6-specific paracrine mechanism [35]. The link between p38 MAPK and IL-6 has also been shown in human cardiac fibroblasts [78].

In addition of increased IL-6 expression and p38 MAPK phosphorylation, we show an increased TNC expression in cardiac fibroblasts of PIEZO1^M-R/M-R^ mice treated with Yoda1. TNC is mostly absent in normal adult heart but re-expressed under pathological conditions, such as dilated cardiomyopathy, myocarditis or myocardial infarction [79,80,81,82]. The major source of TNC in the heart is fibroblasts and this protein is implicated in myofibroblast differentiation [80,83]. It is a well-known profibrotic molecule with genetic deletion of TNC being associated with reduced lung fibrosis [84] and reduced pressure-overload-induced cardiac fibrosis and myocyte hypertrophy [85]. PIEZO1 has been linked to TNC in a recent study with Yoda1 treatment increasing TNC expression in rat cardiac fibroblasts [15]. IL-6 and p38 MAPK are also associated with TNC expression in different systems and notably cardiac fibroblasts [36,86,87,88,89]. Other profibrotic molecules such as collagens have been linked to PIEZO1/IL-6/p38 MAPK cascade with IL-6 treatment of cultured cardiac fibroblasts increasing collagen secretion [90] or PIEZO1 controlling different types of collagen in osteoblastic cells [91].

Similarities in global gene expression, hypertrophy and fibrosis have been shown in 8-week-old mice overexpressing PIEZO1 specifically in cardiomyocytes, with increased heart size, profibrotic collagen content, *ANP* (not *BNP*) and *β-MHC* mRNA levels [23]. However, these mice developed spontaneous dilated cardiomyopathy while we did not observe pump function defects in our model, suggesting a different mechanism involved. Fibroblasts can communicate with neighbouring cardiomyocytes [92] and act on myocyte size, playing a crucial role in the development of cardiomyocyte hypertrophy [70,93]. However, the communication between the two cell types is complex and still unclear. We do not exclude a direct effect of PIEZO1^M-R/M-R^ in cardiomyocytes but, as PIEZO1 levels are twenty times higher in fibroblasts than cardiomyocytes [13], we hypothesise a bigger implication of fibroblasts in the cardiac phenotype observed in PIEZO1^M-R/M-R^ mice. We only studied homozygous mice, but it is notable that about one third of the African population is estimated to carry a PIEZO1 GOF mutation as heterozygotes, causing DHS and malarial protection [32,94,95]. The phenotype described here in PIEZO1^M-R/M-R^ mice might differ depending on the mutation’s zygosity. 

In conclusion, our data suggest that GOF PIEZO1 channels impact the heart, at least partly through amplification of PIEZO1 activity and downstream signalling in cardiac fibroblasts. However, we do not eliminate the possibility that indirect secondary effects of DHS could participate to this cardiac phenotype. It will be useful in the future to investigate the pathological cardiovascular effects of this mutant at an older age and under conditions of stress—e.g., transverse aortic constriction-mediated pressure overload-induced cardiac hypertrophy, fibrosis and HF—to evaluate any beneficial or deleterious effects of PIEZO1 GOF mutation on cardiac function and disease progression, as PIEZO1 appears to be a key protein involved in maintaining homeostatic functional state of the heart. It will also be interesting to determine whether the corresponding M2225R human mutation or other GOF mutations have similar effects in people, and if they underlie a greater tendency for cardiac disease.

## Figures and Tables

**Figure 1 cells-11-01199-f001:**
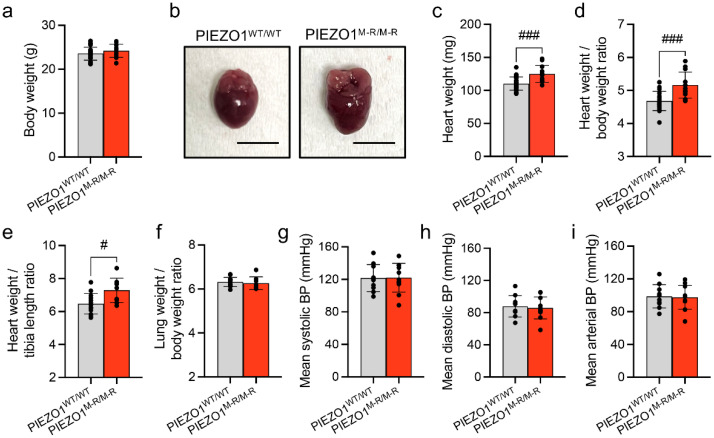
PIEZO1 gain-of-function mutation causes increases heart size without changes in blood pressure. (**a**) Body weight of PIEZO1^WT/WT^ and PIEZO1^M-R/M-R^ mice. (**b**) Representative whole heart images of PIEZO1^WT/WT^ and PIEZO1^M-R/M-R^ mice. Scale bar = 0.5 cm. (**c**–**f**) Morphometric analysis of PIEZO1^WT/WT^ and PIEZO1^M-R/M-R^ mice. (**c**) Heart weight; (**d**) ratio of heart weight to body weight; (**e**) ratio of heart weight to tibia length; (**f**) ratio of lung weight to body weight. (**g**–**i**) Blood pressure (BP) parameters. (**g**) Systolic; (**h**) diastolic; (**i**) mean arterial BP of PIEZO1^WT/WT^ and PIEZO1^M-R/M-R^ mice. Data are for *n* = 22 PIEZO1^WT/WT^ and 12 PIEZO1^M-R/M-R^ mice for (**a**–**d**); *n* = 13 PIEZO1^WT/WT^ and 8 PIEZO1^M-R/M-R^ mice for (**e**,**f**) and *n* = 10 PIEZO1^WT/WT^ and 11 PIEZO1^M-R/M-R^ mice for (**g**–**i**) (mean ± S.D.). Superimposed dots represent the individual values for each animal. # *p* < 0.05; ### *p* < 0.001 vs. PIEZO1^WT/WT^ mice. Unpaired Student’s *t*-test was used to evaluate statistical significance.

**Figure 2 cells-11-01199-f002:**
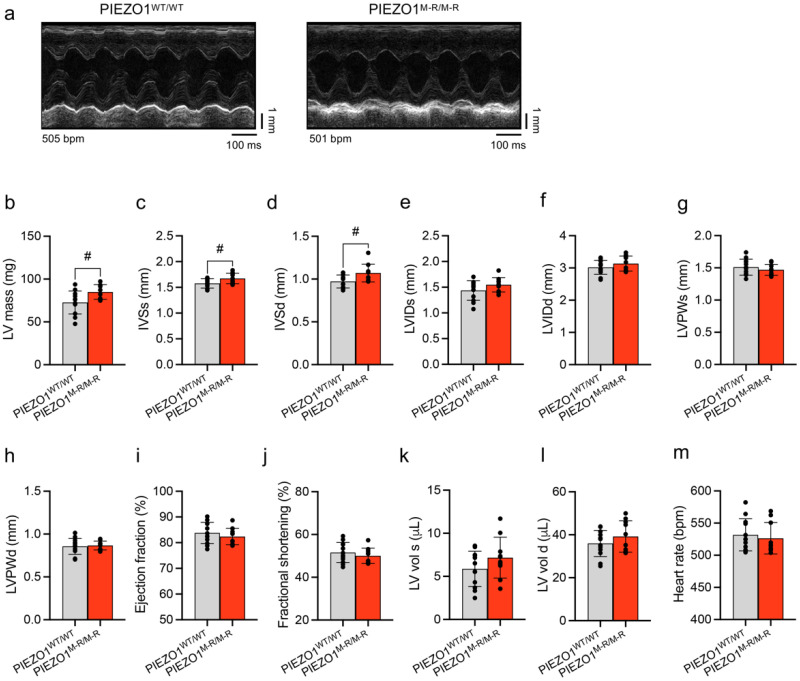
PIEZO1 gain-of-function mutation causes in vivo cardiac hypertrophy. (**a**) Representative echocardiography M-mode images of short axis view in PIEZO1^WT/WT^ and PIEZO1^M-R/M-R^ mice. (**b**–**m**) Parameters obtained from the echocardiograms analysis. (**b**) Left ventricular (LV) mass; (**c**) interventricular septum thickness at end-systole (IVSs); (**d**) IVS in end-diastole (IVSd); (**e**) LV internal diameter in systole (LVIDs); (**f**) LVID in diastole (LVIDd); (**g**) LV posterior wall thickness in systole (LVPWs); (**h**) LVPW in diastole (LVPWd); (**i**) cardiac ejection fraction; (**j**) cardiac fractional shortening; (**k**) LV volume in systole (LV vol s); (**l**) LV vol in diastole (LV vol d); (**m**) heart rate. Data are for *n* = 12 PIEZO1^WT/WT^ and 10 PIEZO1^M-R/M-R^ mice (mean ± S.D.). Superimposed dots represent the individual values for each animal. # *p* < 0.05 vs. PIEZO1^WT/WT^ mice. Unpaired Student’s *t*-test was used to evaluate statistical significance.

**Figure 3 cells-11-01199-f003:**
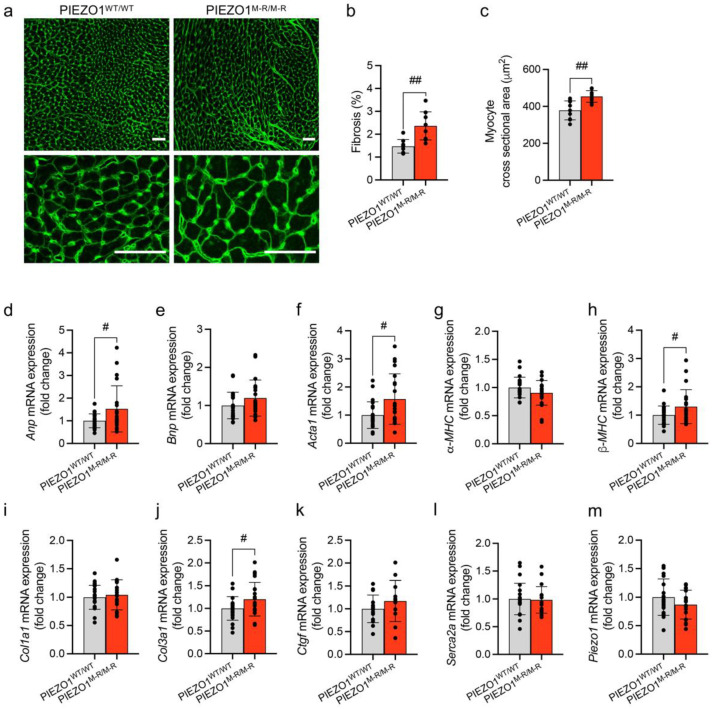
PIEZO1 gain-of-function mutation causes cardiomyocyte hypertrophy and myocardial fibrosis. (**a**) WGA (wheat germ agglutinin) stained heart cross-sections of the PIEZO1^WT/WT^ and PIEZO1^M-R/M-R^ mice. Scale bars = 50 μm. (**b**) Quantitative analysis of total fibrotic area and (**c**) cardiomyocyte cross-sectional area using images of the type shown in (**a**). (**d**–**m**) Quantitative PCR mRNA expression data for *Anp*, *Bnp*, *Acta1*, *α-MHC*, *β-MHC*, *Col1a1*, *Col3a1*, *Ctgf*, *Serca2a* and *Piezo1* genes in whole heart of PIEZO1^WT/WT^ and PIEZO1^M-R/M-R^ mice. mRNA levels were normalised to reference gene expression and presented as fold change relative to PIEZO1^WT/WT^ mice. Data are for *n* = 9 PIEZO1^WT/WT^ and 9 PIEZO1^M-R/M-R^ mice for (**a**–**c**); *n* = 21–23 PIEZO1^WT/WT^ and 18–21 PIEZO1^M-R/M-R^ mice for (**d**–**j**,**l**,**m**); *n* = 15 PIEZO1^WT/WT^ and 12 PIEZO1^M-R/M-R^ mice for (**k**) (mean ± S.D.). Superimposed dots represent the individual values for each animal. # *p* < 0.05; ## *p* < 0.01 vs. PIEZO1^WT/WT^ mice. Unpaired Student’s *t*-test was used to evaluate statistical significance.

**Figure 4 cells-11-01199-f004:**
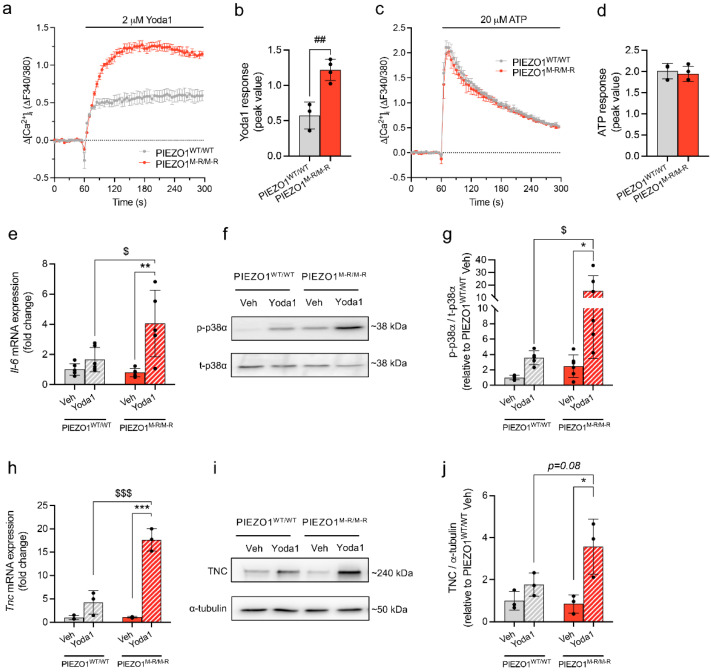
PIEZO1 gain-of-function mutation causes increased agonist-evoked Ca^2+^-entry in cardiac fibroblasts and amplified downstream signalling. (**a**) Examples of changes (∆) in intracellular free Ca^2+^ above baseline in response to 2 μM Yoda1 in cultured fibroblasts isolated from PIEZO1^WT/WT^ or PIEZO1^M-R/M-R^ mice (*n* = 3 replicates per data point). (**b**) Mean data for the type of experiment shown in (**a**), measured 60–90 s after Yoda1 treatment. (**c**) As for (**a**), but 20 μM ATP was used in place of Yoda1. (**d**) Mean data for the type of experiment shown in (**c**), measured 60–90 s after ATP treatment. (**e**) Quantitative PCR mRNA expression data for *Il-6* gene in PIEZO1^WT/WT^ and PIEZO1^M-R/M-R^ fibroblasts exposed to 10 μM Yoda1 or vehicle control (DMSO) for 6 h, normalised to reference gene expression, and presented as fold change relative to PIEZO1^WT/WT^ vehicle control. (**f**) Representative Western blot for phosphorylated p38α protein (p-p38α) and total p38α protein in PIEZO1^WT/WT^ and PIEZO1^M-R/M-R^ fibroblasts exposed to 10 μM Yoda1 or vehicle control only (DMSO) for 10 min. (**g**) Mean data for the type of experiment shown in (**f**), quantification of p-p38α expression relative to total p38α expression in PIEZO1^WT/WT^ and PIEZO1^M-R/M-R^ fibroblasts. (**h**) Quantitative PCR mRNA expression data for *Tnc* gene in PIEZO1^WT/WT^ and PIEZO1^M-R/M-R^ fibroblasts exposed to 10 μM Yoda1 or vehicle control (DMSO) for 6 h, normalised to reference gene expression and presented as fold-change change relative to PIEZO1^WT/WT^ vehicle control. (**i**) Representative Western blot for TNC protein in PIEZO1^WT/WT^ and PIEZO1^M-R/M-R^ fibroblasts exposed to 10 μM Yoda1 or vehicle control only (DMSO) for 6 h, normalised to α-tubulin protein expression. (**j**) Mean data for the type of experiment shown in (**i**), quantification of TNC expression relative to α-tubulin expression in PIEZO1^WT/WT^ and PIEZO1^M-R/M-R^ fibroblasts. Data are for *n* = 3 PIEZO1^WT/WT^ and 4 PIEZO1^M-R/M-R^ mice for (**a**–**d**); *n* = 6 PIEZO1^WT/WT^ and 5 PIEZO1^M-R/M-R^ mice for (**e**); *n* = 5 PIEZO1^WT/WT^ and 6 PIEZO1^M-R/M-R^ mice for (**f**,**g**); *n* = 3 PIEZO1^WT/WT^ and 3 PIEZO1^M-R/M-R^ mice for (**h**–**j**) (mean ± S.D.). Superimposed dots represent the individual values for each animal. ## *p* < 0.01 vs. PIEZO1^WT/WT^ mice. * *p* < 0.05; ** *p* < 0.01; *** *p* < 0.001 vs. PIEZO1^M-R/M-R^ vehicle-treated fibroblasts. $ *p* < 0.05; $$$ *p* < 0.001 vs. PIEZO1^WT/WT^ Yoda1-treated fibroblasts. Statistical significance was evaluated using unpaired Student’s *t*-test in (**b**,**d**); and using two-way ANOVA followed by Tukey’s HSD post hoc test for multiple comparisons in (**e**,**g**,**h**,**j**).

**Figure 5 cells-11-01199-f005:**
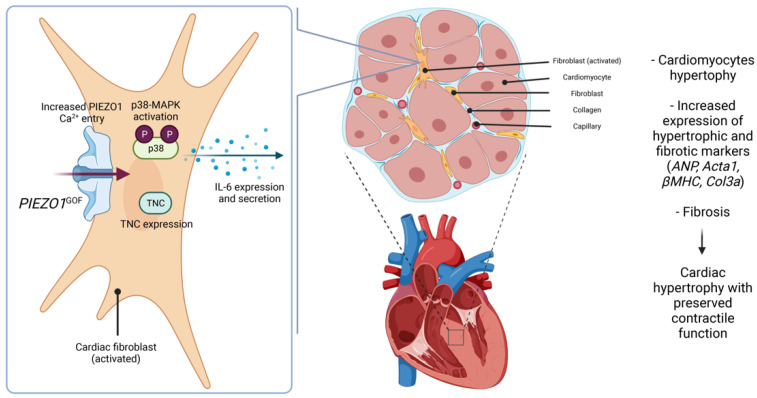
Schematic of proposed mechanism by which PIEZO1 gain-of-function (GOF) mutation causes cardiac remodelling. The M2241R PIEZO1 GOF mutation increased PIEZO1-dependent calcium entry and amplified downstream p38, IL-6 and TNC signalling in cardiac fibroblasts, which correlated to global hypertrophic and fibrotic remodelling of the heart.

**Table 1 cells-11-01199-t001:** List of primers used for RT-qPCR of heart samples.

Gene	Forward Primer	Reverse Primer
*Anp*	AGGCCATATTGGAGCAAATC	CTCCTCCAGGTGGTCTAGCA
*Bnp*	ATGGATCTCCTGAAGGTGCTG	GTGCTGCCTTGAGACCGAA
*Acta1*	CGTGAAGCCTCACTTCCTACC	AGAGCCGTTGTCACACACAA
*α-MHC*	GCCCAGTACCTCCGAAAGTC	GCCTTAACATACTCCTCCTTGTC
*β-MHC*	CTACAGGCCTGGGCTTACCT	TCTCCTTCTCAGACTTCCGC
*Col1a1*	GCTCCTCTTAGGGGCCACT	CCACGTCTCACCATTGGGG
*Col3a1*	CTGTAACATGGAAACTGGGGAAA	CCATAGCTGAACTGAAAACCACC
*Ctgf*	GGGCCTCTTCTGCGATTTC	ATCCAGGCAAGTGCATTGGTA
*Serca2a*	CACACCGCTGAATCTGAC	GGAAGCGGTTACTCCAGT
*Piezo1*	TGAGCCCTTCCCCAACAATAC	CTGCAGGTGGTTCTGGATATAG
*Rpl32*	GCTGCTGATGTGCAACAAA	GGGATTGGTGACTCTGATGG

## Data Availability

The data presented in this study are available from the corresponding authors upon request.

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
