# Peer review of "Global PIEZO1 Gain-of-Function Mutation Causes Cardiac Hypertrophy and Fibrosis in Mice"

_cells, 2022, doi:10.3390/cells11071199_

Round 1
Reviewer 1 Report
Bartoli et al. investigated the effects of global PIEZO1 gain-of-function mutation on the heart and demonstrated that homozygous mutant (M2241R) mice exhibited cardiac hypertrophy and fibrosis without alteration in contractile function. They also showed that mutant-induced elevated Ca2+ entry resulted in p38 MAPK activation and IL-6 overexpression in cultured cardiac fibroblasts and concluded that PIEZO1 gain-of-function mutation causes cardiac remodeling. This study provides the novel finding and important aspects of cardiac remodeling. However, the reviewer has some major concerns before acceptation for publication. Details are listed below.
1) Do the authors hypothesize that mutation M2241R in cardiac cells resulted in the cardiac hypertrophy and fibrosis? If the M2241R mouse is a good model of the human dehydrated hereditary stomatocytosis (DHS) (Line 54-57), is there any possibility that the observed alteration in the murine heart could be the secondary results of anemia? For instance, what about the effects of sympathetic nerve activation and humoral catecholamine?
2) Not only mRNA but also protein levels of some important key molecules should be measured. Plus, collagen production from cultured cardiac fibroblasts should be measured because in vivo results show fibrosis in mutant mice and fibroblasts (and myofibroblasts) are major sources of fibrogenic matrixes. In addition, what is the relationship between collagen production and Ca2+-MAPK-IL6?
3) Line 490-494: Have the author investigated whether this mutant mouse is sensitive to pathological conditions? If so, adding these results would strengthen their conclusion.
4) Although there are citations about the role of PIEZO1 in cardiomyocytes in Introduction sention, the discussions about the present results (increases in heart size and myocyte cross sectional area) should be mentioned in Discussion section. Is there any contribution of fibroblasts to myocyte size? Plus, a schematic diagram showing the main hypothesis of this study will help to summarize the present findings.
Author Response
Bartoli et al. investigated the effects of global PIEZO1 gain-of-function mutation on the heart and demonstrated that homozygous mutant (M2241R) mice exhibited cardiac hypertrophy and fibrosis without alteration in contractile function. They also showed that mutant-induced elevated Ca2+ entry resulted in p38 MAPK activation and IL-6 overexpression in cultured cardiac fibroblasts and concluded that PIEZO1 gain-of-function mutation causes cardiac remodeling. This study provides the novel finding and important aspects of cardiac remodeling. However, the reviewer has some major concerns before acceptation for publication. Details are listed below.
Thank you for your time and feedback on our manuscript. We have made every effort to clarify the manuscript and respond constructively. We have provided a point-by-point responses to your comments below and a highlighted version of the manuscript to show all the changes.
Point 1: Do the authors hypothesize that mutation M2241R in cardiac cells resulted in the cardiac hypertrophy and fibrosis? If the M2241R mouse is a good model of the human dehydrated hereditary stomatocytosis (DHS) (Line 54-57), is there any possibility that the observed alteration in the murine heart could be the secondary results of anemia? For instance, what about the effects of sympathetic nerve activation and humoral catecholamine?
Response 1: We do not eliminate the possibility that indirect secondary effects of DHS could participate to this cardiac phenotype and added a statement to this effect in the last paragraph of the Discussion. Most DHS patients have mild anaemia phenotype and two thirds have fully compensated haemolysis (Picard et al., 2019). We find that the anaemia is also mild in the mouse model at the young age used in this study (Evans et al., 2020). As such, although we do not exclude the possibility, we suspect that the anaemia does not account for the cardiac effects reported here. Moreover, we see a clear effect on cardiac fibroblasts, so there would seem to be a stronger possibility for the effects originating in the heart itself.
We found no reports of increased sympathetic nerve activation in DHS, in contrast to patients with iron deficiency anaemias (A. Hamed et al., 2020). Autonomic nervous system activity during general anaesthesia was not modified in our mice as their heart rate was similar to controls (Figure 2m). Catecholamines play a central role in blood pressure regulation, but we found no differences in blood pressure parameters in PIEZO1M-R/M-R mice compared to controls. We did not further investigate these systems
DHS is associated with iron overload, which can contribute to cardiac effects (Kremastinos and Farmakis, 2011). However, a recent study using a similar PIEZO1 gain-of-function mouse model found iron overload until advanced age (over 1 year old), with young adults (2-5 months old) showing no sign of iron overload (Ma et al., 2021). As we used young adults in our study (8-12 weeks old), we do not expect iron overload to be present and contribute to the cardiac phenotype. Therefore, we hypothesise that the M2241R mutation in cardiac cells is the primary reason for the cardiac phenotype. We show an effect of this mutation on cardiac fibroblasts so hypothesised a downstream effect of this particular cell type on the cardiac phenotype.
Point 2: Not only mRNA but also protein levels of some important key molecules should be measured. Plus, collagen production from cultured cardiac fibroblasts should be measured because in vivo results show fibrosis in mutant mice and fibroblasts (and myofibroblasts) are major sources of fibrogenic matrixes. In addition, what is the relationship between collagen production and Ca2+-MAPK-IL6?
Response 2: Fibroblasts and myofibroblasts are the predominant production sites of collagen in heart (Cowling et al., 2019) and are most likely responsible for the elevated type 3 collagen expression and fibrotic content seen in whole heart tissue. We do not have data to provide for collagen but other ECM proteins are secreted by cardiac fibroblasts such as tenascin-c (Tnc), highly expressed during cardiac impairment and well-known to induce fibrosis (Bhattacharyya et al., 2016; Imanaka-Yoshida and Hiroe, 2009; Kasprzycka et al., 2015). We added new data on Tnc mRNA and protein expression in control and PIEZO1M-R/M-R isolated cardiac fibroblasts treated with Yoda1 agonist or vehicle (Figure 4h-j). We show an upregulation of Tnc mRNA expression and TNC protein expression upon Yoda1 treatment which is amplified in PIEZO1M-R/M-R fibroblasts. These new data support our claim that the amplified PIEZO1 function and IL-6 / p38 / TNC downstream signalling specifically in PIEZO1M-R/MR fibroblasts are contributors for the fibrosis induction.
We extended the discussion on TNC and the relationship between profibrotic molecules and PIEZO1 / IL-6 / p38 MAPK. Page 11, line 536-549: “In addition of increased IL-6 expression and p38 MAPK phosphorylation, we show an increased TNC expression in cardiac fibroblasts of Piezo1M-R/M-R mice treated with Yoda1. TNC is mostly absent in normal adult heart but re-expressed under pathological conditions such as dilated cardiomyopathy, myocarditis, or myocardial infarction (Imanaka-Yoshida et al., 2020, 2004, 2001; Matsumoto and Aoki, 2020). The major source of TNC in the heart is fibroblasts and this protein is implicated in myofibroblast differentiation (Imanaka-Yoshida et al., 2020; Katoh et al., 2020). It is a well-known profibrotic molecule with genetic deletion of TNC being associated with reduced lung fibrosis (Bhattacharyya et al., 2016) and reduced pressure overload-induced cardiac fibrosis and myocyte hypertrophy (Podesser et al., 2018). PIEZO1 has been linked to TNC in a recent study with Yoda1 treatment increasing TNC expression in rat cardiac fibroblasts (Ploeg et al., 2021). IL-6 and p38 MAPK are also associated with TNC expression in different systems and notably cardiac fibroblasts (Fujimoto et al., 2013; Maqbool et al., 2016; Shimojo et al., 2015; Suzuki et al., 2018; Tong et al., 2020). Other profibrotic molecules such as collagens have been linked to PIEZO1 / IL-6 / p38 MAPK cascade with IL-6 treatment of cultured cardiac fibroblasts increasing collagen secretion (Zhang et al., 2016) or PIEZO1 controlling different types of collagen in osteoblastic cells (Wang et al., 2020)”.
Point 3: Line 490-494: Have the author investigated whether this mutant mouse is sensitive to pathological conditions? If so, adding these results would strengthen their conclusion.
Response 3: We did not have the opportunity to look yet at this interesting point but we plan to investigate it in a near future.
Point 4: Although there are citations about the role of PIEZO1 in cardiomyocytes in Introduction sention, the discussions about the present results (increases in heart size and myocyte cross sectional area) should be mentioned in Discussion section. Is there any contribution of fibroblasts to myocyte size? Plus, a schematic diagram showing the main hypothesis of this study will help to summarize the present findings.
Response 4: We briefly mentioned the similarities between our model and the one overexpressing PIEZO1 in cardiomyocytes in the conclusion part, but we moved it to a more appropriate section in the discussion. Page 12, line 550-554: “Similarities in global gene expression, hypertrophy and fibrosis have been shown in 8 weeks-old mice overexpressing PIEZO1 specifically in cardiomyocytes, with increased heart size, profibrotic collagen content, ANP (not BNP) and -MHC mRNA levels (Jiang et al., 2021). However, these mice developed spontaneous dilated cardiomyopathy while we did not observe pump function defects in our model suggesting a different mechanism involved.”
Cardiomyocyte-fibroblast communication is important for cardiac structure and function. Fibroblasts play a crucial role in the development of cardiomyocyte hypertrophy (Fujiu and Nagai, 2014; Hall et al., 2021). Following cardiac injury, cardiac fibroblasts become activated and secrete growth factors and extracellular matrix components which can stimulate cardiac hypertrophy. Cardiomyocytes similarly respond by secreting multiple factors. Fibroblast-myocyte crosstalk is crucial for cardiac disease progression (Hall et al., 2021; Kakkar and Lee, 2010). However, communication between myocytes and fibroblasts is complex and it is still not clear whether fibroblasts directly induce cardiac hypertrophy, without prior signalling from cardiac myocytes. We don’t exclude a direct effect of the M2241R mutation in cardiomyocytes but as PIEZO1 levels are twenty times higher in fibroblasts than cardiomyocytes (Blythe et al., 2019), we hypothesise a bigger implication of fibroblasts in the cardiac phenotype seen in Piezo1M-R/M-R mice.
We also added discussion to clarify and recognise the communication between cardiomyocytes and fibroblasts regarding myocyte hypertrophy and to acknowledge that M2241R mutation in cardiomyocytes could account for the cardiac phenotype exhibited by Piezo1M-R/M-R mice. Page 12, line 557-560: “Fibroblasts can communicate with neighbouring cardiomyocytes (Kakkar and Lee, 2010) and act on myocyte size, playing a crucial role in the development of cardiomyocyte hypertrophy (Fujiu and Nagai, 2014; Hall et al., 2021). However, the communication between the two cell types is complex and still unclear. We do not exclude a direct effect of Piezo1M-R/M-R in cardiomyocytes but as PIEZO1 levels are twenty times higher in fibroblasts than cardiomyocytes (Blythe et al., 2019), we hypothesise a bigger implication of fibroblasts in the cardiac phenotype observed in Piezo1M-R/M-R mice”.
Following your recommendation, we created a graphical abstract showing the main hypothesis and results of our study (see fig 5).
References:
- Hamed, S., F. Elhadad, A., F. Abdel-aal, R., A. Hamed, E., 2020. Cardiac Autonomic Function with Iron Deficiency Anemia. J. Neurol. Exp. Neurosci. 6, 51–57. https://doi.org/10.17756/jnen.2020-075
Bhattacharyya, S., Wang, W., Morales-Nebreda, L., Feng, G., Wu, M., Zhou, X., Lafyatis, R., Lee, J., Hinchcliff, M., Feghali-Bostwick, C., Lakota, K., Budinger, G.R.S., Raparia, K., Tamaki, Z., Varga, J., 2016. Tenascin-C drives persistence of organ fibrosis. Nat. Commun. 7, 11703. https://doi.org/10.1038/ncomms11703
Blythe, N.M., Muraki, K., Ludlow, M.J., Stylianidis, V., Gilbert, H.T.J., Evans, E.L., Cuthbertson, K., Foster, R., Swift, J., Li, J.,
Drinkhill, M.J., van Nieuwenhoven, F.A., Porter, K.E., Beech, D.J., Turner, N.A., 2019. Mechanically activated Piezo1
channels of cardiac fibroblasts stimulate p38 mitogen-activated protein kinase activity and interleukin-6 secretion.
- Biol. Chem. 294, 17395–17408. https://doi.org/10.1074/jbc.RA119.009167
Cowling, R.T., Kupsky, D., Kahn, A.M., Daniels, L.B., Greenberg, B.H., 2019. Mechanisms of cardiac collagen deposition in
experimental models and human disease. Transl. Res. 209, 138–155. https://doi.org/10.1016/j.trsl.2019.03.004
Evans, E.L., Povstyan, O.V., De Vecchis, D., Macrae, F., Lichtenstein, L., Futers, T.S., Parsonage, G., Humphreys, N.E.,
Adamson, A., Kalli, A.C., Ludlow, M.J., Beech, D.J., 2020. RBCs prevent rapid PIEZO1 inactivation and expose slow
deactivation as a mechanism of dehydrated hereditary stomatocytosis. Blood 136, 140–144.
https://doi.org/10.1182/blood.2019004174
Fujimoto, M., Suzuki, H., Shiba, M., Shimojo, N., Imanaka-Yoshida, K., Yoshida, T., Kanamaru, K., Matsushima, S., Taki, W.,
- Tenascin-C induces prolonged constriction of cerebral arteries in rats. Neurobiol. Dis. 55, 104–109.
https://doi.org/10.1016/j.nbd.2013.01.007
Fujiu, K., Nagai, R., 2014. Fibroblast-mediated pathways in cardiac hypertrophy. J. Mol. Cell. Cardiol. 70, 64–73.
https://doi.org/10.1016/j.yjmcc.2014.01.013
Hall, C., Gehmlich, K., Denning, C., Pavlovic, D., 2021. Complex Relationship Between Cardiac Fibroblasts and
Cardiomyocytes in Health and Disease. J. Am. Heart Assoc. 10, e019338. https://doi.org/10.1161/JAHA.120.019338
Imanaka-Yoshida, K., Hiroe, M., 2009. Tenascin-C Regulates Fibrosis, Inflammation and Immunological Response. J. Card.
Fail. 15, S143. https://doi.org/10.1016/j.cardfail.2009.07.199
Imanaka-Yoshida, K., Hiroe, M., Nishikawa, T., Ishiyama, S., Shimojo, T., Ohta, Y., Sakakura, T., Yoshida, T., 2001. Tenascin-
C modulates adhesion of cardiomyocytes to extracellular matrix during tissue remodeling after myocardial
infarction. Lab. Investig. J. Tech. Methods Pathol. 81, 1015–1024. https://doi.org/10.1038/labinvest.3780313
Imanaka-Yoshida, K., Hiroe, M., Yoshida, T., 2004. Interaction between cell and extracellular matrix in heart disease: multiple
roles of tenascin-C in tissue remodeling. Histol. Histopathol. 19, 517–525. https://doi.org/10.14670/HH-19.517
Imanaka-Yoshida, K., Tawara, I., Yoshida, T., 2020. Tenascin-C in cardiac disease: a sophisticated controller of inflammation,
repair, and fibrosis. Am. J. Physiol. Cell Physiol. 319, C781–C796. https://doi.org/10.1152/ajpcell.00353.2020
Jiang, F., Yin, K., Wu, K., Zhang, M., Wang, S., Cheng, H., Zhou, Z., Xiao, B., 2021. The mechanosensitive Piezo1 channel
mediates heart mechano-chemo transduction. Nat. Commun. 12, 869. https://doi.org/10.1038/s41467-021-21178-4
Kakkar, R., Lee, R.T., 2010. Intramyocardial Fibroblast Myocyte Communication. Circ. Res. 106, 47–57.
https://doi.org/10.1161/CIRCRESAHA.109.207456
Kasprzycka, M., Hammarström, C., Haraldsen, G., 2015. Tenascins in fibrotic disorders—from bench to bedside. Cell Adhes.
Migr. 9, 83–89. https://doi.org/10.4161/19336918.2014.994901
Katoh, D., Kozuka, Y., Noro, A., Ogawa, T., Imanaka-Yoshida, K., Yoshida, T., 2020. Tenascin-C Induces Phenotypic Changes
in Fibroblasts to Myofibroblasts with High Contractility through the Integrin αvβ1/Transforming Growth Factor
β/SMAD Signaling Axis in Human Breast Cancer. Am. J. Pathol. 190, 2123–2135.
https://doi.org/10.1016/j.ajpath.2020.06.008
Kremastinos, D.T., Farmakis, D., 2011. Iron Overload Cardiomyopathy in Clinical Practice. Circulation 124, 2253–2263.
https://doi.org/10.1161/CIRCULATIONAHA.111.050773
Ma, S., Dubin, A.E., Zhang, Y., Mousavi, S.A.R., Wang, Y., Coombs, A.M., Loud, M., Andolfo, I., Patapoutian, A., 2021. A role
of PIEZO1 in iron metabolism in mice and humans. Cell 184, 969-982.e13. https://doi.org/10.1016/j.cell.2021.01.024
Maqbool, A., Spary, E.J., Manfield, I.W., Ruhmann, M., Zuliani-Alvarez, L., Gamboa-Esteves, F.O., Porter, K.E., Drinkhill, M.J.,
Midwood, K.S., Turner, N.A., 2016. Tenascin C upregulates interleukin-6 expression in human cardiac
myofibroblasts via toll-like receptor 4. World J. Cardiol. 8, 340. https://doi.org/10.4330/wjc.v8.i5.340
Matsumoto, K.-I., Aoki, H., 2020. The Roles of Tenascins in Cardiovascular, Inflammatory, and Heritable Connective Tissue
Diseases. Front. Immunol. 11, 609752. https://doi.org/10.3389/fimmu.2020.609752
Picard, V., Guitton, C., Thuret, I., Rose, C., Bendelac, L., Ghazal, K., Aguilar-Martinez, P., Badens, C., Barro, C., Bénéteau, C.,
Berger, C., Cathébras, P., Deconinck, E., Delaunay, J., Durand, J.-M., Firah, N., Galactéros, F., Godeau, B., Jaïs, X., de
Jaureguiberry, J.-P., Le Stradic, C., Lifermann, F., Maffre, R., Morin, G., Perrin, J., Proulle, V., Ruivard, M., Toutain,
F., Lahary, A., Garçon, L., 2019. Clinical and biological features in PIEZO1 -hereditary xerocytosis and Gardos
channelopathy: a retrospective series of 126 patients. Haematologica 104, 1554–1564.
https://doi.org/10.3324/haematol.2018.205328
Ploeg, M.C., Munts, C., Prinzen, F.W., Turner, N.A., van Bilsen, M., van Nieuwenhoven, F.A., 2021. Piezo1 Mechanosensitive
Ion Channel Mediates Stretch-Induced Nppb Expression in Adult Rat Cardiac Fibroblasts. Cells 10, 1745.
https://doi.org/10.3390/cells10071745
Podesser, B.K., Kreibich, M., Dzilic, E., Santer, D., Förster, L., Trojanek, S., Abraham, D., Krššák, M., Klein, K.U., Tretter, E.V.,
Kaun, C., Wojta, J., Kapeller, B., Gonçalves, I.F., Trescher, K., Kiss, A., 2018. Tenascin-C promotes chronic pressure
overload-induced cardiac dysfunction, hypertrophy and myocardial fibrosis. J. Hypertens. 36, 847–856.
https://doi.org/10.1097/HJH.0000000000001628
Shimojo, N., Hashizume, R., Kanayama, K., Hara, M., Suzuki, Y., Nishioka, T., Hiroe, M., Yoshida, T., Imanaka-Yoshida, K.,
- Tenascin-C May Accelerate Cardiac Fibrosis by Activating Macrophages via the Integrin αVβ3/Nuclear
Factor–κB/Interleukin-6 Axis. Hypertension 66, 757–766. https://doi.org/10.1161/HYPERTENSIONAHA.115.06004
Suzuki, T., Muraki, Y., Hatano, N., Suzuki, H., Muraki, K., 2018. PIEZO1 Channel Is a Potential Regulator of Synovial Sarcoma
Cell-Viability. Int. J. Mol. Sci. 19, 1452. https://doi.org/10.3390/ijms19051452
Tong, X., Zhang, Jinjing, Shen, M., Zhang, Junyang, 2020. Silencing of Tenascin-C Inhibited Inflammation and Apoptosis Via
PI3K/Akt/NF-κB Signaling Pathway in Subarachnoid Hemorrhage Cell Model. J. Stroke Cerebrovasc. Dis. Off. J.
Natl. Stroke Assoc. 29, 104485. https://doi.org/10.1016/j.jstrokecerebrovasdis.2019.104485
Wang, L., You, X., Lotinun, S., Zhang, L., Wu, N., Zou, W., 2020. Mechanical sensing protein PIEZO1 regulates bone
homeostasis via osteoblast-osteoclast crosstalk. Nat. Commun. 11, 282. https://doi.org/10.1038/s41467-019-14146-6
Zhang, Y., Wang, J.-H., Zhang, Y.-Y., Wang, Y.-Z., Wang, J., Zhao, Y., Jin, X.-X., Xue, G.-L., Li, P.-H., Sun, Y.-L., Huang, Q.-H.,
Song, X.-T., Zhang, Z.-R., Gao, X., Yang, B.-F., Du, Z.-M., Pan, Z.-W., 2016. Deletion of interleukin-6 alleviated
interstitial fibrosis in streptozotocin-induced diabetic cardiomyopathy of mice through affecting TGFβ1 and miR-29
pathways. Sci. Rep. 6, 23010. https://doi.org/10.1038/srep23010
Reviewer 2 Report
The manuscript by Bartoli et al. studies the effect of mechanosensing molecule PIEZO in concern of cardiac remodeling. The author showed a detrimental effect of cardiac function and remodeling with the gain of function of PIEZO. Overall, the work is novel and can be considered for the readership of Cells after improving it.
- This manuscript has some data to show an association between the dynamics of inflammation and It would be more appropriate if the authors could include some introductory parts in the introduction section that discuss how inflammation dynamics and PIEZO are co-related.
- The authors showed pro-inflammatory cytokine IL-6 affected with Yoda1. It would be more appropriate if the Authors could analyze the dynamics of proinflammatory and anti-inflammatory cytokines including TGFb to get a broader overview to depict that it is specific to IL-6 or inflammation dynamics.
Round 2
Reviewer 1 Report
The revised version has been improved according to the reviewer’s comments. Especially, the answers to points numbered 1 and 2 were constitutive and convincing. On the other hand, the manuscript should be revised also according to points No. 3 and 4.
No. 3:
The authors only answered “No” to the comment No. 3. The revised version should include their future plan about this matter.
No.4:
The figure should be added to the manuscript.
Author Response
Response to Reviewer 1 Comments
The revised version has been improved according to the reviewer’s comments. Especially, the answers to points numbered 1 and 2 were constitutive and convincing. On the other hand, the manuscript should be revised also according to points No. 3 and 4.
Thank you for your positive feedback and comments on our manuscript. We responded constructively to the additionnal points you’ve raised. We have provided a point-by-point responses to your comments below and a highlighted version of the manuscript to show all the changes.
Point 3: The authors only answered “No” to the comment No. 3. The revised version should include their future plan about this matter.
Response 3: We briefly mentioned our future plans in the conclusion part and now added more details about this matter. Page 11, line 539-544: “It will be useful in the future to investigate the pathological cardiovascular effects of this mutant at an older age and under conditions of stress; e.g. transverse aortic constriction-mediated pressure overload-induced cardiac hypertrophy, fibrosis and HF; to evaluate any beneficial or deleterious effects of PIEZO1 GOF mutation on cardiac function and disease progression, as PIEZO1 appears to be a key protein involved in maintaining homeostatic functional state of the heart.
Point 4: The figure should be added to the manuscript.
Response 4: As graphical abstracts are not shown in the main text and are published separately, we now include the schematic to the manuscript as Figure 5 for more clarity and visibility.